# The Composition and Antioxidant Activity of Bound Phenolics in Three Legumes, and Their Metabolism and Bioaccessibility of Gastrointestinal Tract

**DOI:** 10.3390/foods9121816

**Published:** 2020-12-07

**Authors:** Liuying Zhu, Wenting Li, Zeyuan Deng, Hongyan Li, Bing Zhang

**Affiliations:** 1State Key Laboratory of Food Science and Technology, Nanchang University, Nanchang 330047, Jiangxi, China; 402337519034@email.ncu.edu.cn (L.Z.); 407213315069@email.ncu.edu.cn (W.L.); dengzy@ncu.edu.cn (Z.D.); lihongyan@ncu.edu.cn (H.L.); 2Institute for Advanced Study, University of Nanchang, Nanchang 330031, Jiangxi, China

**Keywords:** soybean, vicia faba, kidney bean, acid/alkaline hydrolysis, in vitro digestion, colonic fermentation

## Abstract

The composition and antioxidant activity of bound phenolics in three legumes (soybean, vicia faba, and kidney bean), and their metabolism and bioaccessibility in the gastrointestinal tract were investigated in this study. The total phenolic content, total flavonoid content, and antioxidant activities (ABTS and FRAP) were evaluated. The phytochemical compositions of the three legumes after acid/alkaline hydrolysis, simulated gastrointestinal digestion, and colonic fermentation were identified and quantified by UPLC-ESI-QTOF-MS/MS and HPLC-ESI-QqQ-MS/MS. The results showed that the three legumes were rich in bound phenolic compounds, and possessed a strong antioxidant activity; among which kidney bean showed a higher bound flavonoid content and antioxidant activity than the other two legumes. Alkaline hydrolysis allowed a more thorough extraction of the bound phenolics of the three legumes than acid hydrolysis. The released contents of bound phenolics were extremely low in in vitro digestion, whereas colonic fermentation favored the release of more phenolic compounds. Kidney bean, which presented the highest bound flavonoid content and antioxidant activity, had the lowest bioaccessibility. Our study provides a wider insight into the constituents and bioavailability of bound phenolic compounds in the three legumes.

## 1. Introduction

Plant polyphenols are a kind of secondary metabolite, which are widely found in various foods, such as legumes, fruits, vegetables, herbs, and cereals [1]. They are important bioactive components in our diet. Many studies associated polyphenol-rich diets with several health effects in human, and triggered the increasing interest in polyphenol. The phenolics present in plants mainly include two parts, soluble phenols that can be extracted by organic solvents, and insoluble bound phenols that are not extracted by organic solvents [2]; the bound phenolic mainly bonds with cellulose, hemicellulose, pectin, protein, and arabinoxylans by ester and C–C bonds in cell walls, acting as building materials for the cell wall matrix, so they cannot be extracted by the organic solvent and present in the residue [3,4]. Nevertheless, the chemical bonds between the substances can be destroyed by acid hydrolysis, alkaline hydrolysis, or enzymatic hydrolysis [5], thus releasing bound phenolic substances. At present, most research on phenolic substances are concentrated on the soluble phenols, while research on bound phenolic has been neglected, resulting in the underestimation of the total content of plant polyphenols, their antioxidant capacity, and their effects on the human body.

In recent years, legumes as a human diet, have been winning growing interest for their health effects, and are recommended as great source of bioactive phenolic compounds, which constitute plenty of biological functions [6]. Several studies have demonstrated that legumes possess numerous bioactive properties, which have been largely connected with their phenolic compounds. García-Lafuente et al. [7] showed that phenolic extract of common bean can stimulate the expression of cytokine mRNA in macrophages, and reduce the production of nitrous oxide, indicating that it has high antioxidant and anti-inflammatory activities. It has been reported that there are abundant phenolic acids (delphinidin and ferulic acid) and anthocyanins in beans, which are usually applied as functional food ingredients, and may be beneficial to health, such as being anti-tumor, and prevention of cardiovascular disease [8]. Additionally, phenolic compounds in legumes can control postprandial glucose response by inhibiting α-glucosidase to reduce the digestion and absorption of glucose in the intestine, suggesting a use for the management of type 2 diabetes mellitus [9]. In short, these results clarify the bioaccessibility of the phenolic compounds in legumes. 

As widely known, the health benefits of phenolic compounds are closely related to metabolism [10]. Mosele et al. [11] reported the absorption of soluble polyphenols mainly in the small intestine, while bound polyphenols are metabolized by the intestinal flora in the colon, instead of being absorbed in the small intestine [12]. Therefore, it is of great significance to study the chemical transformation of bound polyphenols during gastrointestinal digestion and colonic fermentation. The multiple advantages of in vitro research, such as being fast and cheap, better condition control, and less ethical restrictions [13], have made it an alternative method to studying the metabolic process of biologically active substances in vivo. 

Although legumes are considered a potential source of soluble and bound phenols, there is no information on the bioaccessibility of bound phenols during gastrointestinal digestion and subsequent colonic fermentation. Soybean, vicia faba, and kidney bean are common edible beans, and rich in phenolic components [6]. Hence the present study screened these three legumes to examine their bound phenolic composition and antioxidant activities, as well as their metabolism and bioaccessibility of bound phenolics by in vitro gastrointestinal digestion and colon fermentation (anaerobically incubated with human feces).

## 2. Materials and Methods

### 2.1. Materials 

Soybean (*Glycine max* (Linn.) Merr.), vicia faba (*Vicia faba* L.), and kidney bean (*Phaseolus vulgaris L*.) were obtained from the local supermarket in Nanchang, Jiangxi, China (February 2016). The three legumes were freeze-dried, milled to a fine powder with a high-speed universal pulverizer and sieved through 20 meshes, then kept in a sealed bag at −80 °C until extracted. 

### 2.2. Chemicals and Reagents

P-hydroxybenzoic acid, p-coumaric acid, chlorogenic acid, ferulic acid, procatechuic acid, gallic acid, sinapic acid, catechin, daidzein, daidzin, quercetin, hyperoside, genistein, genistin, rutin, naringenin, glycitein, glycitin, biochanin A, and vitexin as phenolic standards were bought from Aladdin (Shanghai, China). Na_2_CO_3_, FeCl_3_∙6H_2_O, CaCl_2,_ NaCl, KCl, KH_2_PO_4_, and K_2_HPO_4_, NaHCO_3_, MgSO_4_H_2_O, L-cysteine, K_2_S_2_O_8,_ FeSO_4_, HCl, and NaOH were obtained from Xilong (Guangzhou, China). Acetate buffer, PBS buffer, Tween resazurin solution, arabinogalactan, xylan, α-amylase, trypsin, bile salt, Trolox, Folin–Ciocalteu, 2,2′-azinobis-(3-ethylbenzthiazoline-6-sulphonate) (ABTS), and 1,3,5-tri(2-pyridyl)-2,4,6-triazine (TPTZ) were bought from Sigma (St. Louis, MO, USA). HPLC-grade solvents, including methanol and formic acid, were obtained from Merck (Darmstadt, Germany). The water used in this work was ultrapure water produced by a Milli-Q system.

### 2.3. Extraction of Bound Phenolic Compounds

The samples (1 g) of the three legumes were weighed in a 50 mL centrifuge tube to extract free phenolic compounds in an ultrasonic bath at room temperature with 15 mL 80% (*v/v*) aqueous methanol for 1 h, then centrifuged (4200× *g*) for 10 min to collect the supernatant. The samples were extracted four times, and the methanol extract was free phenolic compounds. The residues were used to extract bound phenolic compounds according to the methods of Peng Han and Kim, K.-H, with slight modifications [14,15]. The bound phenolics obtained by alkaline hydrolysis were hydrolyzed at room temperature with 20 mL NaOH (2 M) under nitrogen for 4 h; the pH of the solution was brought to 2 with 6 M HCl, and then centrifuged for 10 min (4200× *g*) to collect the supernatant; residues were re-extracted five times. Acid hydrolysis of the samples was conducted with 25 mL HCl (2 M) at 85 °C for 1 h; then the pH of the solution was adjusted to 2 using 10 M NaOH and centrifuged (4200× *g*, 10 min) to collect the supernatant. Finally, the supernatants obtained by acid or alkaline hydrolysis were evaporated and resolved with 5 mL 80% methanol, and 1 mL was filtered through a 0.22 µm PTFE membrane for HPLC analyses. The samples were preserved at −20 °C in a refrigerator for further analysis. 

### 2.4. Gastrointestinal Digestion of Bound Phenolic 

The in vitro gastrointestinal (GI) digestion was conducted with the sample residues after extraction of soluble phenolic by organic solvents, and the experiment was carried out by a published method with a slightly modification [13]. First, oral digestion was simulated: 3 g of residue was blended with 2.1 mL of simulated saliva digestive fluid (SSF) electrolyte stock solution, and 0.3 mL α-amylase (1500 U/mL) and 15 μL CaCl_2_ (0.3 M) were added, then the mixture was diluted with 585 mL water, and the mixture oscillated at 37 °C in a water bath for 5 min. Second, gastric digestion was simulated: 4.5 mL of simulated gastric digestive fluid (SGF) electrolyte stock solution, 0.96 mL pepsin stock solution (25,000 U/mL), and 3 μL CaCl_2_ (0.3 M) were added to the above oral digestive system, and the pH of the mixture was adjusted to 3 with 0.12 mL HCl (1 M); then 0.417 mL water was added and the solution was incubated at 37 °C for 2 h. Finally, intestinal digestion was simulated: 6.6 mL simulated intestinal digestive fluid (SIF) electrolyte stock solution, 3 mL trypsin (800 U/mL), 1.5 mL fresh bile (160 mM), and 24 μL CaCl_2_ (0.3 M) were blended in 12 mL gastric chime; then the mixture was regulated to pH 7.0 with 0.15 mL NaOH (1 M), 1.31 mL water was added, and kept at 37 °C for 2 h, the GI digestion was centrifuged (4200× *g*, 10 min) to collect supernatant, and preserved at −20 °C until analysis.

### 2.5. In Vitro Colonic Fermentation

A previously reported method was used to conduct in vitro fermentation with some appropriate modifications [16], and the specific operation process of in vitro fermentation was undertaken as follows.

#### 2.5.1. In Vitro Fermentation Growth Medium Preparation

Growth medium (1 L) was composed of 4.5 g NaCl and KCl, 0.5 g KH_2_PO_4_ and K_2_HPO_4_,1.5 g NaHCO_3_, 0.7 g MgSO_4_H_2_O, 0.8 g L-cysteine HCl·H_2_O, 0.005 g FeSO_4_·7H_2_O, 0.08 g CaCl_2_, 0.4 g bile salt, Tween 80 (1 mL), resazurin solution (4 mL, 0.025%, *w/v*), 2 g arabinogalactan, and 1 g xylan. Before preparing the sample, the prepared medium was sterilized in glass vessels at 121 °C for 15 min. 

#### 2.5.2. Fecal Extract Preparation

The residues collected from the in vitro GI digestion were fermented in vitro by human gut microbiota, which was obtained from three healthy donors (22–28 years) without antibiotic treatment for 3 months before experimentation, and without a history of gastrointestinal diseases; a two-day polyphenol free diet was conducted before sample collection. All healthy donors provided written informed consent. Samples were preserved in an anaerobic tank immediately after collection, and then diluted with 10% (*v/v*) PBS buffer and blended to acquire a suspension embodying 10% (*w/v*) phosphate buffer. The suspensions were used as the fermentation initiator.

#### 2.5.3. Fermentation Conditions 

Fermentation broth (10 mL) consisted of 45% fecal suspensions, 45% growth medium, and 10% legume residues collected from the in vitro GI digestion. Fermentation was started by mixing suspensions, fecal suspensions, and legume samples and incubated at 37 °C at 200 strokes /min for 48 h. The samples were collected at 0, 1, 3, 6, 12, 24, 36, and 48 h, centrifuged, and 1 ml was filtered through a 0.22 µm PTFE membrane for HPLC analyses. The pH value of the fermentation broth was detected at each period. The samples were saved at −80 °C in a refrigerator until analysis.

### 2.6. Determination of Total Phenolic Content (TPC) and Total Flavonoid Content (TFC)

#### 2.6.1. Determination of TPC

A Folin–Ciocalteu assay was used to determine the TPC, as described in previous methods [17]. In short, 25 μL samples or gallic acid standards (5, 25, 50, 100, 200, 400, 600 μg/mL) were blended with Folin–Ciocalteu reagent (125 μL, 0.2 M) in a 96 well microplate and incubated in an incubator at 37 °C for 6 min; 125 μL Na_2_CO_3_ (10 g/100 mL) solution was added and reacted for 30 min. An ELX800 microplate reader (BioTek Instruments, Inc., Winooski, VT, USA) was used to read the absorbance at 765 nm. The TPC was calculated as milligrams of gallic acid equivalent (GAE, mg GAE/g DW).

#### 2.6.2. Determination of TFC 

An Erinitrit aluminum trichloride assay was used to determine the TFC [17]. In brief, 25 μL of catechin standard (5, 25, 50, 100, 200 μg/mL) or bean extract was mixed with 110 μL NaNO_2_ (0.066 M) in a 96-well microplate and kept for 5 min at room temperature; 15 μL AlCl_3_ (0.75 M) was added and reacted at room temperature for 6 min; then 100 μL NaOH (0.5 M) solution was added and reacted at 37 °C for another 10 min. An ELX800 microplate reader (BioTek Instruments, Inc., Winooski, VT, USA) was used to read the absorbance of the mixture at 510 nm. The TFC was calculated as milligram catechin equivalent (CAE, mg CAE/g DW).

### 2.7. Antioxidant Activity

#### 2.7.1. ABTS Radical Scavenging Activity

ABTS was assessed based on a previously reported method, with slight modifications [18]. ABTS working solution was prepared through blending 0.2 mL ABTS^+^ stock solution (7.4 mM) with 0.2 mL K_2_S_2_O_8_ (2.6 mM) in a ratio of 1:1 (*v/v*), and reacting for 12 h in the dark. The ABTS working solution was diluted with 80% ethanol to obtain the absorbance of 0.70 ± 0.05 at 734 nm. Then, 200 μL of ABTS working solution was mixed with 10 μL of samples or Trolox standards (20, 40, 50, 60, 70, 80 μg/mL) and incubated in a 96 well plate for 6 min in the dark. A Thermo Varioskan Flash Microplate Reader (Thermo Scientific, Waltham, MA, USA) was used to read the absorbance at 734 nm. All samples were measured three times. The antioxidant activity was denoted as micromole Trolox equivalents (TE) per gram of dry weight (DW) (mmol TE/g DW).

#### 2.7.2. Ferric Reducing Antioxidant Power (FRAP Assay)

For FRAP assay, the procedure was based on the method of Zhang B et al. [17]. Acetate buffer (300 mM), FeCl_3_∙6H_2_O (20 mM), and 10 mM TPTZ in 40 mM HCl were mixed at a ratio of 10:1:1 *(v/v/v*) and warmed to 37 °C to prepare the fresh working solution. Then, 5 μL of samples or FeSO_4_ standards (0.03125, 0.0625, 0.125, 0.25, 0.5, 1 mmol/L) was reacted with 180 μL working solution in a 96 well plate for 10 min at 37 °C. A Thermo Varioskan Flash Microplate (Thermo Scientific, Waltham, MA, USA) was used to measure the absorbance at 593  nm. The antioxidant activity was denoted as FeSO_4_ equivalents (FE, mmol FE/g DW).

### 2.8. Qualitative Analysis by UPLC-ESI-QTOF-MS/MS 

#### 2.8.1. Liquid Cromatographic Conditions

Each of the samples was detected by a 1290 infinity series Ultra Performance Liquid Chromatography (UPLC) system (Agilent Technologies, Santa Clara, CA, USA) equipped with a diode array detector (DAD) and Agilent Eclipse Plus C18 column (2.1 mm × 100 mm, 1.8 μm), with a column heater set to 35 °C. The mobile phase was composed of 0.1% formic acid in de-ionized water (solvent A) and methanol (solvent B), and passed at a flow rate of 0.3 mL/min The linear gradient was as follows: 0–10 min, 5–10% B; 10–30 min, 10–30% B; 30–38 min, 30–50% B; 38–43 min, 50% B; 43–45 min, 50–5% B. The UV–visible absorbance of the peak was monitored between 200 and 400 nm. The injection volume was 5 µL. 

#### 2.8.2. Mass Spectrometric Conditions

An orthogonal acceleration quadrupole time-of-flight mass spectrometer (Agilent Technologies, Santa Clara, CA, USA) equipped with an Electron Spray Ionization (ESI) source was used for qualitative analysis. Phenolic compounds were monitored by the negative ion mode in a *m*/*z* range from 50 to 1000. The mass capillary voltage was kept at 4.5 kV and the flow rate of drying gas (N_2_) was 10 L/min, with the temperature at 300 °C. Nebulizer pressure, fragmentor voltage, and collision energy were set at 30 psi, 175 V, and 10–40 eV, respectively. 

### 2.9. Quantitative Analysis by HPLC-ESI-QqQ-MS

#### 2.9.1. LC-MS Conditions

The samples were quantified by HPLC mass spectrometry (MS) analysis. The sample was injected into an Agilent Eclipse Plus C18 column (2.1 mm × 100 mm, 1.8 μm). Then, 0.1% formic acid in de-ionized water (solvent A) and methanol (solvent B) were used as mobile phase. The elution gradient was as follows: 0 min, 5% B; 18 min, 50% B; 25 min, 95% B; 28 min, 5% B. Other HPLC conditions were the same as for UPLC. 

A triple quadrupole mass spectrometer (Agilent Technologies, Santa Clara, CA, USA) equipped with an ESI source was used for quantitative analysis. The instrument was utilized in the negative mode, and in multiple reactions monitoring (MRM) mode. The mass capillary voltage was kept at +4.0 kV and the drying gas flow rate was 11 L/min with the temperature at 300 °C. Nebulizer pressure was set at 15 psi, and the fragment voltage and collision energy of each material was optimized.

#### 2.9.2. Calibration and Quantification of Phenolics

A calibration curve was used to evaluate the linearity for 21 phenolic standards, which were subjected to accurate weighing (1 mg), and dissolved in methanol (1 mL), then serially diluted to 10, 25, 50, 100, 150, and 200 µg/mL. 

### 2.10. Statistical Analysis

All experiments were performed in triplicate, and the data were presented as mean value ± standard deviations (SD). The limit of detection (LOD) and limit of quantitation (LOQ) were determined from the signal to noise ratio (S/R), considering the LOD and LOQ to be the lowest concentration capable of generating an S/R ≥ 3 and S/R ≥ 10, respectively. The data were analyzed by one-way analysis of variance (ANOVA), and Duncan’s multiple range test was used to determine statistical differences on the level of significance at *P <* 0.05. All statistical analyses were conducted using SPSS statistical software (Version 18.0, SPSS Inc., Armonk, New York, NY, USA). 

## 3. Result and Discussion

### 3.1. Total Phenolics and Total Flavonoids of Legumes

Total phenolic contents and total flavonoid contents of bound phenolics released by acid/alkaline hydrolysis in the different legumes are presented in (Table 1). Results show that the TPC value of the three legumes, obtained by acid hydrolysis treatment, was between 0.012 and 0.31 mg GAE/g DW, while the TPCs obtained by the alkali hydrolysis treatment of the three legumes were remarkably higher than those by acid hydrolysis (1.79–2.27 mg GAE/g DW), indicating that alkaline hydrolysis was more efficient in releasing the bound phenolic compounds than acid hydrolysis, which is in accordance with earlier studies [19]. Moreover, the TPC of kidney bean (2.07 ± 0.09 mg GAE/g DW), released through alkaline hydrolysis, was slightly lower than that of soybean (2.27 ± 0.30 mg GAE/g DW), but without statistically significant difference. Interestingly, the soybean exhibited the lowest TPC of acid hydrolysis bound phenolics, only 0.012 ± 0.001 mg GAE/g DW. In addition, compared with the free phenolics (Appendix A), there was no significant difference in TPC between bound phenolics (1.79–2.27 mg GAE/g DW) and free phenolics (2.04–2.48 mg GAE/g DW), which means that the main component of phenols in the three legumes were not only free phenolics, but that bound phenolics also occupied a large part. Wang et al. [20] also reported the contents of bound phenolics in 14 legumes, including soybeans, broad beans, and kidney beans, and ranging from 32.6% to 68.3%.

Kim et al. [15] reported that the release of phenolic compounds under alkaline hydrolysis is better than that under acid hydrolysis. This may be due to alkaline hydrolysis being able to cleave the ester bonds between phenolic acid and polysaccharide, and reduce the loss of phenolic acids [19,21]. Among the three legumes, kidney beans have attracted more and more attention because of their excellent source of dietary antioxidants. Kan et al. [22] reported the TPC values of 26 kidney bean seed coats were between 0.25 and 35.11 mg GAE/g DW. In this study, kidney bean contained dramatically higher TPC and TFC than the other two legumes, indicating that kidney bean is also an excellent source of bound phenolic.

### 3.2. Identification and Quantification of Bound Phenolic Compounds of Three Legumes in Acid/Alkaline Hydrolysis

The base peak chromatograms (BPC) obtained from the three legumes extracts are presented in (Figure 1). There were 9, 8, and 13 phenolic compounds released by acid hydrolysis in soybean, vicia faba, and kidney bean, while the phenolic compounds released with alkali hydrolysis were 12, 18, and 15 in soybean, vicia faba, and kidney bean, being 25%, 55%, and 13% superior, respectively. This result further confirmed that alkali hydrolysis was more effective than acid hydrolysis in releasing phenolic compounds from the three legumes. 

The retention times, MS^2^ fragmentation pattern in negative, and UV spectra are shown in (Table 2). In summary, 36 bound phenolic were identified by comparison with the relevant standards, literature reports, or databases. The identified compounds can be classified into three groups, including phenolic acids and their derivatives, flavonoids and their derivatives, and other compounds. As seen in Table 2, 17 phenolic compounds, 9 flavonol compounds, and 10 other compounds were identified in the three legumes, among which 10 phenolic compounds were already identified in the soluble phenolic compounds (Appendix A), including protocatechuic acid, p-coumaric acid, dihydroxybenzoic acid, catechin, ferulic acid, sinapic acid, daidzin, genistin, vitexin, and quercetin.

#### 3.2.1. Phenolic Acids 

A total of 17 phenolic acids and their derivatives were authenticated in the three legumes, most of which were observed in both kidney bean and vicia faba, but only compounds 3, 9, 24, 25, 29, and 35 were found in soybean. As shown in Table 2, compounds 1 (t_R_ 3.555 min, *m*/*z* 169), 3 (t_R_ 5.503 min, *m/z* 153), 14 (t_R_ 15.614 min, *m*/*z* 193), 16 (t_R_ 18.010 min, *m*/*z* 223), 24 (t_R_ 23.762 min, *m*/*z* 353), and 25 (t_R_ 24.728 min, *m/z* 163) were authenticated as gallic acid, protocatechuic acid, ferulic acid, sinapic acid, chlorogenic acid, and p-coumaric acid, respectively, through comparison with the retention time and mass spectrum data of the respective standard compounds. Compound 19 (t_R_ 21.018 min, *m/z* 223) and compound 16 (t_R_ 18.010 min, *m*/*z* 223) were tentatively authenticated as the isomer of sinapic acid, due to the same molecular ion but a different retention time. Compound 2 (t_R_ 3.623 min, *m*/*z* 197), with a fragment ion at *m/z* 153 [M-H-CO^2^]^-^, corresponded to syringic acid [23]. Compound 4 (t_R_ 6.860 min, *m/z* 153), exhibiting fragment ions at *m/z* 109 [M-H-CO^2^]^−^, was characterized as dihydroxybenzoic acid [24]. Compound 5 (t_R_ 7.168 min, *m/z* 181) was regarded as hydroxyphenyllactic acid, due to its fragment ions at *m/z* 153.0190 [M-H-CO]^−^, 135 [M-H-CO-H_2_O]^-^, and 162 [M-H-H_2_O]^−^, and by referring to the Metlin online database. Based on a reported study [14], compounds 6, 7, and 9 (t_R_ 9.110, 9.996 and 11.181 min), which displayed the same [M-H]^−^ ion at *m/z* 137, were preliminarily characterized as hydroxybenzoic acid. Compound 18 (t_R_ 20.836 min, *m/z* 167) revealed a major fragment ion at *m/z* 148.8648[M-H-H_2_O]^−^, which was authenticated as 4-hydroxyphenylglycolic acid [25]. Compounds 22 and 35 (t_R_ 22.890 and 36.650 min, *m/z* 163) were initially regarded as coumaric acid based on a literature comparison [26]. Compound 29 (t_R_ 33.099 min, *m/z* 343) was authenticated as a coumaric acid derivative, classified according to its characteristic fragment ion at *m/z* 163.

#### 3.2.2. Flavonoids 

A total of nine flavonoids, including isoflavones, flavones, flavonols, flavanones, flavanes, and their derivatives were observed in the three legumes, among which compounds 27 (daidzin), 31 (genistin), and 32 (vitexin) were only found in soybean, compound 36 (quercetin), 21 (myricetin), 26 (ampelopsin), and 30 (hovenitin I) were only present in kidney bean, compound 30 was only characterized in vicia faba, while compound 13 (catechin) was found in both kidney bean and vicia faba. 

Compounds 13 (t_R_ 15.484 min, *m/z* 289), 27 (t_R_ 28.416 min, *m/z* 415), 31 (t_R_ 33.690 min, *m/z* 431), 32 (t_R_ 34.043 min, *m/z* 431) and 36 (t_R_ 41.448 min, *m/z* 301) were characterized as catechin, daidzin, genistin, vitexin, and quercetin, respectively, with an authentic standard. Compounds 20 (t_R_ 20.956 min, *m/z* 269), 21 (t_R_ 22.425 min, *m/z* 317), 26 (t_R_ 27.935 min, *m/z* 319), and 30 (t_R_ 33.285 min, *m/z* 333) were preliminarily authenticated as trihydroxyflavone, myricetin, ampelopsin, and hovenitin I, respectively, by comparison with other works [14,23]. Under this condition, some non-phenolic substances were also detected. After comparing with the Metlin online database, compounds 8, 15, and 17 (t_R_ 11.134, 16.830, 18.357 min, *m/z* 153), with the same [M-H]^-^ ion at *m/z* 153 and typical fragment ions at 123[M-H-CH_2_O]^−^ and 125[M-H-CO]^−^, were tentatively classified as hydroxytyrosol, whereas compound 10 (t_R_ 12.667 min, *m/z* 161), compound 28 (t_R_ 31.256 min, *m/z* 217), and compound 33 (t_R_ 36.732 min, *m/z* 287) were regarded as 4-hydroxycoumarin, eupatoriochromene, and 2-hydroxyestradiol, respectively. Compound 11 (t_R_ 13.223 min) carried the same molecular ion at *m/z* 121, but its retention time was earlier than compound 12 (t_R_ 14.701 min), and was identified as m-hydroxybenzaldehyde, and compound 12 as p-hydroxybenzaldehyde [27]. Compound 23 (t_R_ 23.010 min, *m/z* 239) displayed a fragment ion at *m/z* 195, suggesting that it could be alizari [28]. Compound 34 (t_R_ 35.801 min, *m/z* 261) carried a fragment ion at *m/z* 125 and 187, and was characterized as maclurin, according to previously reported data [29]. 

The results showed that the kind of bound phenolic compounds found differed greatly in soybean, vicia faba, and kidney bean. More specifically, compounds 9, 24, 27, 31, 32, and 34 (Hydroxybenzoic acid, chlorogenic acid, daidzin, genistin, vitexin, and maclurin, respectively) were only detected in the hydrolysates of soybean, while compounds 2, 19, 20, 8, 10, 23, and 28 (syringic acid, sinapic acid, trihydroxyflavone, hydroxytyrosol, 4-hydroxycoumarin, alizarin, and eupatoriochromene, respectively) were only present in the hydrolysates of vicia faba, and compounds 6, 16, 18, 36, 21, 26, 30, 11, and 33 (Hydroxybenzoic acid, sinapic acid, 4-hydroxyphenylglycolic acid, quercetin, myricetin, ampelopsin, hovenitin I, and 2-hydroxyestradiol, respectively) were only found in the hydrolysates of kidney bean. The type of bound phenol compound released by hydrolysis was related to the method of hydrolysis, in addition to the kind of legume. There were differences in the composition of different methods of extracting bound phenolic compounds; some phenolic substances could only be detected from acid hydrolysates or alkali hydrolysates, such as compounds 2, 5, 14, 16, 19, 24, 25, 20, 32, 8, 10, and 33 (syringic acid, hydroxyphenyl lactic acid, ferulic acid, sinapic acid, chlorogenic acid, p-coumaric acid trihydroxyflavone, vitexin, hydroxytyrosol, 4-hydroxycoumarin, eupatoriochromene, and 2-hydroxyestradiol, respectively) were only found in alkaline hydrolysis, whereas compounds 26, 30, and 11 (Ampelopsin, hovenitin I, and m-hydroxybenzaldehyde, respectively) were only present in acid hydrolysis. This might be due to alkaline hydrolysis being able to efficiently hydrolyze the ether bond or ester bond between phenolic compounds and food substrates, while acid hydrolysis prefers to hydrolyze glycosidic bonds [30,31]. Moreover, the phenolic extracts after alkaline hydrolysis and acid hydrolysis were almost all individual phenols, and with few glycoside phenols, while the soluble phenols have many types of glycoside phenols (Appendix A). 

#### 3.2.3. Quantitative Analysis of Bound Phenolic Compounds 

Triple quadrupole mass spectrometer and external standard methods were used to determine the content of bound phenolic compounds in the legumes. Due to the difficulty of purchasing the standard substances, only 20 chemical components, including 7 phenolic acids and 13 flavonoids were analyzed. 

As shown in (Table 3), protocatechuic acid (7.83–46.87 μg/g DW) was the major phenolic acid in the three legumes, followed by p-hydroxybenzoic acid (0.20–20.74 μg/g DW), and p-coumaric acid (0.66–3.84 μg/g DW). More specifically, the content of gallic acid was highest in kidney bean (18.58 μg/g DW in acid hydrolysis and 9.57 μg/g DW in alkaline hydrolysis), but it was found in small amounts in the acid-hydrolyzed vicia faba (0.32 μg/g DW). Chlorogenic acid was only found in the alkaline hydrolyzed product of soybean (0.84 μg/g DW). Moreover, the phenolic acids produced by acid hydrolysis were obviously higher than by alkaline hydrolysis in kidney bean, among which the protocatechuic acid, sinapic acid, and gallic acid obtained by acid hydrolysis (16.13, 0.2 and 18.58 μg/g DW, respectively) were two times as much as that of alkaline hydrolysis (7.83, 0.08 and 9.57 μg/g DW, respectively), while the content of p-coumaric acid in acid hydrolysis (2.13 μg/g DW) was three times that of alkaline hydrolysis (0.8 μg/g DW), and the p-hydroxybenzoic acid and ferulic acid released by acid hydrolysis (2.14 and 0.96 μg/g DW respectively) were as much as 10 times that of alkaline hydrolysis (0.2 and 0.19 μg/g DW respectively). This result indicated that the phenolic acids in kidney bean mentioned above may be released mainly by breaking glycosidic bonds and solubilizing sugar.

As for the flavonoid contents, isoflavones were mainly detected in soybeans; daidzin and genistin were the main isoflavones in soybean (ranging from 3.54 to 4.17 μg/g DW). For flavones, hyperoside was not measured in the three legumes, and quercetin showed a highest concentration at 4.13 μg/g DW in the acid hydrolysis of vicia faba. More specifically, rutin was only found in the alkaline hydrolysates of the three legumes. As for flavanes, catechin showed the highest concentration at 51.59 μg/g DW in the alkaline hydrolysis of kidney bean, followed by the alkaline hydrolysis of vicia faba (17.54 μg/g DW), however, it was not found in the soybean, indicating that rutin and catechin are mainly covalently bonded to the cell wall by ester bonds and ether bonds.

### 3.3. The Antioxidant Activities of Legumes

The antioxidant activities of bound phenolic compounds in the three legumes were studied by ABTS and FRAP in vitro (Table 1). As indicated, the ABTS value and FRAP value of insoluble-bound phenolic fractions in the three legumes ranged from 1.11 to 3.13 mg TE/g DW, and 4.57 to 30.77 mmol FE/g DW, respectively. Similar to the trend of TPC, the alkaline hydrolysis treatment showed remarkably higher ABTS and FRAP in vicia faba and kidney bean when compared with acid hydrolysis treatment, further verifying that alkaline hydrolysis is a better method of releasing bound antioxidants from these two legumes. However, soybeans showed the opposite trend to kidney beans and vicia faba, the acid hydrolysis treatment exhibited higher ABTS and FRAP (1.54 ± 0.04 mg TE/g DW, 5.86 ± 0.31 mg TE/g DW, respectively) in soybeans when compared to alkaline hydrolysis treatment (1.11 ± 0.09 mg TE/g DW, 4.57 ± 0.25 mg TE/g DW, respectively), indicating that acid hydrolysis is more efficient in releasing the bound antioxidants from soybean. This may be caused by the different chemical components of the scavenging activity. The effectiveness of phenolic compounds extracted from plant-based foods as antioxidants is frequently different [19]. It has reported that the antioxidant activity does not always depend on the quantities of the constituents, but their chemical nature, because considerable variances existed in the efficacies of compounds [32]. In addition, kidney bean had the highest antioxidant activity in the two antioxidant tests, which coincides with the above results, that is, kidney bean showed the highest TPC and TFC. 

It is well known that food rich in phenolics and flavonoids possess excellent antioxidant properties. In this study, Pearson’s correlation coefficient (*R*^2^) was used to analyze the correlation between the phenolic contents (TPC and TFC) and antioxidant activities (ABTS and FRAP) (Appendix A). The results showed that TFC was significantly correlated with ABTS (*R*^2^ = 0.929) and FRAP (*R*^2^ = 0.977), while TPC was weakly correlated with ABTS (*R*^2^ = 0.414) and FRAP (*R*^2^ = 0.455). It can be seen from Table 3 that a high concentration of catechins contributes dramatically to the content of total flavonoids in kidney beans and broad beans, which indicates that flavonoids, especially concentrated catechins, are the main contributors to the reducibility of kidney beans and vicia faba. Meanwhile, these considerable positive correlations indicate that flavonoids are the main contributors to the antioxidant activities of the three legumes, rather than the phenolic compounds. This result is consistent with previous research results that TFC and FRAP in black soybean seed coat extract also had a strong positive correlation [17]. 

### 3.4. The Effect of In Vitro Gastrointestinal Digestion on Bound Phenolics in Three Legumes 

The quantitative results of the bound phenolics after digestion are shown in Table 4. Compared with acid and alkaline hydrolysis treatment, the release of phenolics was extremely low after digestion. For soybean, the total phenolic content released was 2.09 ± 0.23 μg/g DW, 2.80 ± 0.54 μg/g DW, and 3.80 ± 0.36 μg/g DW after simulated oral, gastric, and intestinal digestion, respectively, while for vicia faba it increased to 2.08 ± 0.40 μg/g DW, 2.8 ± 0.72 μg/g DW, and 5.05 ± 0.12 μg/g DW, respectively, and in kidney bean increased to 2.99 ± 0.23 μg/g DW, 4.40 ± 0.59 μg/g DW, and 7.75 ± 0.59 μg/g DW, respectively. More importantly, the content of phenolics produced by intestinal digestion was higher than that produced by oral and stomach digestion in the three legumes. The release of phenolics in the simulated digestion process was about 3.25–14.63% of the alkaline hydrolysis treatment, and 3.65–16.75% of the acid hydrolysis treatment, among which simulated intestinal digestion showed a higher ratio (16.75% and 14.63%, respectively) than oral and stomach digestion. In addition, the phenolic compounds released during the simulated digestion process were different in the three legumes. As for soybean, isoflavones were the main species of phenolic compounds, whereas phenolic acids and flavanes (catechin) were mainly released in vicia faba and kidney bean. More specifically, daidzein, daidzin, genistin, and glycitein were the main compounds released after simulated digestion in soybeans; p-hydroxybenzoic acid, p-coumaric acid, and catechin were released in vicia faba; and ferulic acid, p-hydroxybenzoic acid, procatechuic acid, and catechin were released in kidney bean.

During simulated gastrointestinal digestion, enzymes and pH conditions can promote the hydrolysis of macromolecules [33]. In this study, intestinal digestion was more effective than oral and gastric digestion in releasing bound phenolics, which was attributed to the abundance of hydrolases in the small intestine. Phenolic compounds may be bound to proteins, so they can be released in the enzyme digestion, and become more stable under the pH conditions [34]. The results may show that some phenolic acids and flavonoids in these three legumes were bound to the proteins, and such bound phenolics were varied in the different kinds of beans.

Moreover, some bound phenolic compounds are mainly covalent binding with cell wall substances, but due to the inability of the cell wall fibrous material to be digested, such bound phenolic compounds are retained in the gastric and small intestine, and finally arrive at the colon to be released through the action of several microorganisms and hydrolytic enzymes. Kroon et al. [35] proposed that over 95% of ferulic acid in wheat was released by the enzymatic action (esterase and xylanase activity) of the microbial community during the fermentation in the colon. However, the gastric and small intestinal treatments only released a small amount of ferulic acid (2.6%). Shahidi and Judong [36] showed that phenolic compounds are rarely produced from the food matrix, and are absorbed in the small intestine (5–10%) as the free form, and the remainder move to the colon (large intestine) where the bound phenolics are released by the gut microbiota. 

### 3.5. The Effect of Colonic Fermentation by Human Microflora on Bound Phenolics in Three Legumes 

Phenolics in the diet are difficult to absorb in the small intestine and reach the colon, where they are transformed and degraded by microorganisms. Therefore, in vitro colon fermentation of the bound phenolics in three legumes was carried out. The pH value and the release of bound phenolics of the three legumes during different fermentation times were detected (Appendix A, Table 5). Moreover, the compositions of bound phenolics in the three legumes during different fermentations were identified. The total ion chromatogram of phenolic extracts in the three legumes after colonic fermentation is shown in Figure 2. The retention times and MS^2^ data of identified phenolics in three legumes after colonic fermentation are shown in (Table 6).

#### 3.5.1. The Change of pH During Colonic Fermentation

Colon microbes have a high catabolic activity. They can degrade the components in legume residues, and then change the acidity and alkalinity of the solution [37]. The changes of pH in soybean, broad bean, and kidney bean at different fermentation times are shown in Appendix A. With the increase of fermentation time, the pH values of soybean, vicia faba, and kidney bean were decreased. Interestingly, the pH value of soybean decreased slowly, and the change range within 48 hours was only 0.64, which was significantly lower than the downtrend of vicia faba (2.91) and kidney bean (2.69). It may be that soybean is rich in fat and protein, while vicia faba and kidney bean belong to the high starch beans, and their starch content is significantly higher than soybean. Starch fermentation produces lactic acid and other substances, which can cause the pH of vicia faba and kidney bean to decrease during colon fermentation. Moreover, the pH of vicia faba and kidney bean decreased slowly in 1 h, but decreased significantly in 1–12 h, then tended to be stable after 36 h. Many studies have suggested that only a few gut microbiota (e.g., *Lactobacillus* sp., *Escherichia coli*, *Bacteroides* sp., *Bifidobacterium* sp.,) can biotransform dietary polyphenols, and the fermentation products are mainly phenylpropionic acid and phenylacetic acid or its hydroxyl derivatives [38,39], which can decrease the pH of the gut, and therefore the effect of microorganisms in the early stage of food reaching the colon is more significant. 

#### 3.5.2. Total Phenolic Contents

The content of polyphenols in soybean, vicia faba, and kidney bean at different times of fermentation are shown in Table 5. As the fermentation process proceeded, the polyphenol content of vicia faba increased and then tended to stabilize (0.85–1.45 mg GAE/g DW). This showed that the bound phenolic substances in the legume residues were continuously released by the action of microorganisms and microbial enzymes. The content of polyphenols in soybeans changed with the increase of fermentation time, but showed an increasing trend as a whole. The content of polyphenols in kidney bean was the lowest at 3 h (0.4 mg GAE/g DW), showed an increasing trend after 3 h (from 0.4 to 0.82 mg GAE/g DW), and tended to be stable after 24 h. This may be because the bound phenols were continuously released by microorganisms and microbial enzymes, but at the same time the free phenols were degraded by the microorganisms, so the phenolic acid content was reduced. 

Comparing with the alkaline hydrolysis and in vitro gastrointestinal digestion, the liberative polyphenols of soybean and vicia faba during colonic fermentation were higher, reaching 70.5% and 80.0% of the alkaline hydrolysis bound phenolic content, respectively, while the liberative polyphenols of kidney bean were extremely low, and were only 38.6% of alkaline hydrolysis bound phenolic content. In addition, compared with the soluble polyphenols, the content of phenolic acid released in the fermentation of soybeans, vicia faba, and kidney bean accounted for 84.3%, 69.1%, and 28.9% of the soluble polyphenols (Appendix A), respectively. The present results imply that the liberative and absorptive effects in vivo of bound polyphenols in soybean and vicia faba were significantly higher than kidney bean. In this study, it was found that the TPC, TFC, and antioxidant activity of kidney beans, among the three legumes, was the highest, but the bioaccessibility in the gastrointestinal tract was the lowest, which indicates that even if the TPC, TFC, and antioxidant activity measured in vitro are high, it does not mean that there is high bioaccessibility in the human body. Similarly, it indicates that the utilization results of bound phenols in vivo may not be consistent with those in vitro. 

At present, in vitro fermentation of phenolic compounds is mostly concentrated on extractable phenolic compounds, and few studies have paid attention to the in vitro colonic fermentation of bound phenolic compounds. The concentration of chlorogenic acid, naringin, and rutin standards significantly decreased under the action of intestinal micro-organisms during in vitro colon fermentation [40]. Dall’Asta et al. [41] carried out in vitro colon fermentation of 16 kinds of food juices rich in polyphenols. It was found that the type of phenolic substances changed significantly, and most of the phenolic substances degraded into other phenolic substances after being fermented by microorganisms. Importantly, this study also showed the same results, that changes in the composition of the compounds during different fermentation times indicated that the bound phenolic components were degraded during fermentation in vitro. Under the action of microorganisms, the bound phenolic compounds released during the fermentation of the colon are absorbed by the body in the form of phenolic prototypes or metabolites.

#### 3.5.3. Phenolics

The total ion current chromatogram and preliminary identification of the compounds extracted in the three legume residues at different times of fermentation are shown in Figure 2 and Table 6, respectively. The species of compounds in the extracts changed with different fermentation times. More specifically, 11 species of compounds were identified in kidney bean residue, while 6 species were in vicia faba residue, and 4 species were in soybean residue.

For kidney bean, after comparing with the Metlin online database, compounds 1 (t_R_ 4.635 min, *m/z* 259), 4 (t_R_ 11.852 min, *m/z* 243), 6 (t_R_ 15.591 min, *m/z* 243), 8 (t_R_ 17.986 min, *m/z* 195), and 9 (t_R_ 18.832 min, *m/z* 277) were identified as 2,6,4′-trihydroxy-4-methoxybenzophenone, xanthotoxol acette, benzophenone, acetosyringone, and decyloxy benzoic acid, respectively. Compound 5 (t_R_ 13.820 min, *m/z* 243) showed the same molecular ion and fragment ions at *m/z* 109, with compound 4 was also identified as xanthotoxol acette; compound 2 (t_R_ 8.305 min, *m/z* 229) and compound 11 (t_R_ 48.601 min, *m/z* 391) were characterized as (Iso)pentenyl-7-hydroxy-coumarin and 6-hydroxy kaempferol 3,6,7-trimethyl ether derivative, respectively, by their fragment ions from loss of CO_2_ (44 Da). Compound 3 (t_R_ 10.142 min) with a molecular ion at *m/z* 137 was suggested to be hydroxybenzoic acid according to the literature [42]. Brevifolin carboxylic acid derivative (compound 7, t_R_ 16.069 min) displayed [M-H]^−^ at *m/z* 327 along with characteristic fragment ions at *m/z* 291; compound 10 (t_R_ 44.099 min) with [M-H]^−^ at *m/z* 391, with a fragment ion at *m/z* 223 (sinapic acid), was classified as a sinapic acid derivative [43]. 

For vicia faba, (Iso) pentenyl-7-hydroxy-coumarin (compound 4, t_R_ 8.099 min), sinapic acid derivative (compound 5, t_R_ 43.049 min), and 6-hydroxycalanol 3,6,7-trimethyl ether derivative (compound 6, t_R_ 48.614 min) were also found in the vicia faba. Compound 1 (t_R_ 5.277 min, *m/z* 149), compound 2 (t_R_ 7.996 min, *m/z* 163), and compound 3 (t_R_ 8.001 min, *m/z* 181) were identified as 4-ethylbenzoic acid, hydroxyphenyl acrylate, and dihydroxybenzene propionic acid, respectively, by comparing with the Metlin online database.

In soybean, (Iso) pentenyl-7-hydroxy-coumarin (compound 2, t_R_ 8.103) was also detected. Based on the Metlin online database comparison, compound 1 (t_R_ 3.097 min, *m/z* 217) exhibiting fragment ions at *m/z* 187 [M-H-CO_2_]^−^ and *m/z* 130 [M-H-C_3_H_2_O_3_]^−^ was characterized as 1-hydroxyindole. Compound 3 (t_R_ 11.621 min) exhibited [M-H]- ions at *m/z* 151 and fragments at *m/z* 113 and *m/z* 109 were suggested as hydroxyphenylacetic acid according to the literature data [37]. Compound 4 (t_R_ = 47.147 min) presented a [M-H]^−^ at *m/z* 389, yielding fragments at *m/z* 227 (by loss of 162), suggesting that it could be a resveratrol glucoside [44].

The gut microbiota conducts metabolic reactions to decorate phenolic skeletons, to absorb a range of lower-weight metabolites [45]. The phenols in the fermentation system change with the fermentation time due to microbial enzymes that can make phenolics dehydroxyl, ring fission, and chemical bond breakage, and degrade them to other substances [40]. In this study, sinapic acid derivative and 6-hydroxy kaempferol 3,6,7-trimethyl ether derivative mainly existed in the early stage of fermentation of kidney bean and vicia faba residue, and acetosyringone also existed in the early stage of fermentation of kidney bean residue. This indicated that the released phenolic compounds were being rapidly used or degraded into other metabolites by the gut microbiota. For the components of released by bound polyphenols, hydrolysis and fermentation treatment were different. The bound phenolics released by acid or base hydrolysis were mostly individual phenols in the present study, but fermentation mainly produced phenylpropionic acid, phenylacebic acid, and benzoic acid derivatives (end products), which is consistent with the findings of Rechner A R et al. [40]. Moreover, sinapic acid, coumarin, and kaempferol derivatives, which were found in acid or base hydrolysis, were also present in colonic fermentation. A previous study showed that dietary polyphenols were extensively metabolized to simple phenolics (such as propionic acid and phenylacetic acid derivatives) by the colon, and these compounds can be the biomarkers of colonic metabolism [40]. In this study, phenolic acids such as decyloxy benzoic acid, hydroxybenzoic acid, 4-ethylbenzoic acid, and hydroxyphenylacetic acid were found in colon fermentation. This indicated that phenolic derivatives, such as sinapic acid and kaempferol derivatives, may be degraded to simpler phenolics during colonic fermentation. 

## 4. Conclusions

The legumes were rich in phenolic compounds and possessed strong antioxidant activity, among which kidney bean showed the highest flavonoid contents and antioxidant activity, but a lower bioavailability than the other two legumes, indicating that the utilization results of bound phenol in vivo may not be consistent with that in vitro. Meanwhile, we found that the antioxidant activity of the three legumes depends on the flavonoid compounds, but not the phenolic compounds. Our results also showed that alkaline hydrolysis was more effective than acid hydrolysis in the release of the bound phenolics of the three legumes. In addition, the bound phenolics were released in extremely low quantities in the in vitro simulated digestion (in oral, stomach, and small intestine digestion), only 3.25–14.63% of alkali hydrolysis, but were effectively released from the legume matrix under the action of microorganisms (39.61–82.68% of alkali hydrolysis). The present study also indicated that the products released by chemical hydrolysis and in vitro digestion were different; the main products of chemical hydrolysis are phenolic acids (such as protocatechuic acid), which are metabolized into simple phenolics (such as benzoic acid and phenylacetic acid) after colonic fermentation. The results of this study may help us to better understand the phenolic compounds and their biological utilizations in the soybean, vicia faba, and kidney bean, especially the bound phenolics. 

## Figures and Tables

**Figure 1 foods-09-01816-f001:**
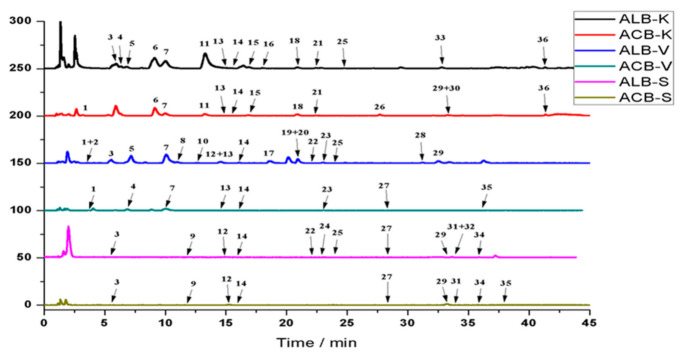
Base peak chromatogram (BPC) of bound phenolics in legumes. ACB-S: acid hydrolysis-bound phenolics of soybean; ALB-S: alkaline hydrolysis-bound phenolics of soybean; ACB-V: acid hydrolysis-bound phenolics of vicia faba; ALB-V: alkaline hydrolysis-bound phenolics of vicia faba; ACB-K: acid hydrolysis-bound phenolics of kidney bean; ALB-K: alkaline hydrolysis-bound phenolics of kidney bean.

**Figure 2 foods-09-01816-f002:**
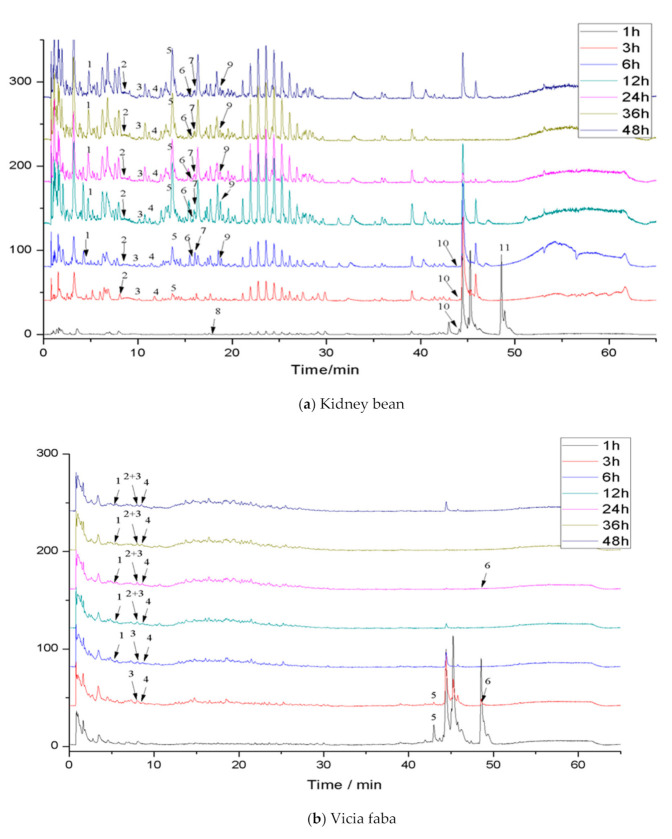
Total ion chromatogram (TIC) of phenolic extracts in legumes after colonic fermentation.

**Table 1 foods-09-01816-t001:** Total phenolic contents, total flavonoids content, and antioxidant activities of bound phenolics in legumes ^A^.

Antioxidative Assay	Hydrolysis Method	Soybean	Vicia Faba	Kidney Bean
TPC	acid	0.012 ± 0.001 ^c^	0.18 ± 0.01 ^b^	0.31 ± 0.01 ^a^
(mg GAE/g DW)	alkaline	2.27 ± 0.30 ^a^	1.79 ± 0.12 ^b^	2.07 ± 0.09 ^ab^
TFC	acid	0.006 ± 0.001 ^c^	0.09 ± 0.03 ^b^	0.17 ± 0.02 ^a^
(mg CAE/g DW)	alkaline	0.13 ± 0.03 ^c^	0.35 ± 0.05 ^b^	0.76 ± 0.03 ^a^
ABTS	acid	1.54 ± 0.04 ^a^	1.11 ± 0.06 ^b^	1.52 ± 0.04 ^a^
(mg TE/g DW)	alkaline	1.11 ± 0.09 ^c^	1.79 ± 0.36 ^b^	3.13 ± 0.11 ^a^
FRAP	acid	5.86 ± 0.31 ^c^	7.58 ± 0.07 ^b^	11.17 ± 0.04 ^a^
(mmol FE/g DW)	alkaline	4.57 ± 0.25^c^	15.24 ± 1.10 ^b^	30.77 ± 1.77 ^a^

TPC, Total Phenolic Content; TFC, Total Flavonoid Content; ABTS, 2,2′-azinobis-(3-ethylbenzthiazoline-6-sulphonate; FRAP, Ferric Reducing Antioxidant Power. ^A^ Results are expressed as mean ± standard deviation of three replicates (*n =* 3); Values followed by the different letters (a, b, c) within the same line are significantly different (*P <* 0.05).

**Table 2 foods-09-01816-t002:** Characterization of the bound phenolic constituents of legumes by UPLC-ESI-QTOF-MS2.

Peak	t_R_	λ_max_	Formula	[M-H]^-^ (*m*/*z*)	Major Fragment Ions (*m*/*z*)	Identification	Source
No.	(min)	(nm)					
**Phenolic acids and derivatives**							
1	3.555	206,273	C_7_H_6_O_5_	169.0148	125.0205[M-H-CO_2_]^−^	Gallic acid ^abc^	V1, V2, K1
2	3.623	206,273	C_9_H_10_O_5_	197.006	153.0227[M-H-CO_2_]^−^	Syringic acid ^ab^	V2
3	5.503	206,273	C_7_H_6_O_4_	153.0189	109.0261[M-H-CO_2_]^−^	Procatechuic acid ^abc^	S1, S2, V2, K1, K2
					125.0214[M-H-CO]^−^		
4	6.86	261,292	C_7_H_6_O_4_	153.0191	109.0286[M-H-CO_2_]^−^	Dihydroxybenzoic acid ^ab^	V1, K2
5	7.168	291,208	C_9_H_10_O_4_	181.012	153.0190[M-H-CO]^−^	Hydroxyphenyllactic acid ^b^	V2, K2
					135.0087[M-H-CO-H_2_O]^−^		
					162.9992[M-H-H_2_O]^−^		
6	9.11	280,311	C_7_H_6_O_3_	137.0222	109.0461[M-H-CO]^−^	Hydroxybenzoic acid ^a^	K1, K2
7	9.996	280,310	C_7_H_6_O_3_	137.0225	119.0129[M-H-H_2_O]^−^	Hydroxybenzoic acid ^a^	V1, V2, K2, K2
					108.0238[M-H-CHO]^−^		
9	11.181	256	C_7_H_6_O_3_	137.0254	119.0115[M-H-H_2_O]^−^	Hydroxybenzoic acid ^a^	S1, S2
					108.0205[M-H-CHO]^−^		
14	15.614	270	C_10_H_10_O_4_	193.0144	133.2188 [M-H-C_2_H_4_O_2_]^−^	Ferulic acid ^abc^	V2, K2
16	18.01	268	C_11_H_12_O_5_	223.021		Sinapic acid ^bc^	K2
18	20.836	264,294	C_8_H_8_O_4_	167.0326	148.8648[M-H-H_2_O]^−^	4-Hydroxyphenylglycolic acid ^a^	K1, K2
19	21.018	273	C_11_H_12_O_5_	223.0358	179.0437[M-H-CO_2_]^−^	Sinapic acid ^ac^	V2
22	22.89	296	C_9_H_8_O_3_	163.0358	119.0500[M-H-CO_2_]^−^	Coumaric acid ^ab^	S2, V2
24	23.762	290	C_16_H_18_O_9_	353.1136	190.8992[M-H-C_6_H_10_O_5_]^−^	Chlorogenic acid ^abc^	S2
25	24.728	290	C_9_H_8_O_3_	163.0402	119.0494[M-H -CO_2_]^−^	p-Coumaric acid ^abc^	S2, V2, K2
29	33.099	271		343.2033	297.1303,163.0575	Coumaric acid derivative ^a^	S1, S2, V2, K1
35	36.65	268	C_9_H_8_O_3_	162.8382	119.0492 [M-H-CO_2_]^−^	Coumaric acid ^ab^	S1,V1
**Flavonoids**							
Isoflavones and derivatives							
27	28.416	282	C_21_H_20_O_9_	415.1493	253.0198[M-C_6_H_10_O_5_]^−^	Daidzin ^abc^	S1, S2
31	33.69	274	C_21_H_20_O_10_	431.0939	269.0404[M-H-C_6_H_10_O_5_]^−^	Genistin ^ac^	S1, S2
Flavones and derivatives							
20	20.956	280	C_15_H_10_O_5_	269.064		Trihydroxyflavone ^a^	V2
32	34.043	298	C_21_H_20_O_10_	431.093	477.1000[M-H+HCOOH]^−^	Vitexin ^abc^	S2
36	41.448	265	C_15_H_10_O_7_	301.0318		Quercetin ^abc^	K1, K2
Flavonols and derivatives							
21	22.425	294	C_15_H_10_O_8_	317.0303	190.9986[M-H-C_6_H_6_O_3_]^−^,	Myricetin ^ab^	K1, K2
					163.0008[M-H-C_7_H_6_O_4_]^−^		
Flavanones and derivatives							
26	27.935	267	C_15_H_12_O_8_	319.04	183.027	Ampelopsin ^a^	K1
30	33.285	268	C_16_H_14_O_8_	333.1053		Hovenitin I ^a^	K1
Flavanes and derivatives							
13	15.484	278	C_15_H_14_O_6_	289.0666	109.1011[M-H-C_9_H_12_O_4_]^−^	Catechin ^abc^	V2, V1, K1, K2
**Other compounds**							
8	11.134	208,273	C_8_H_10_O_3_	153.0183	123.0441[M-H-CH_2_O]^−^,	Hydroxytyrosol ^b^	V2
					125.6422 [M-H-CO]^−^		
10	12.667	285	C_9_H_6_O_3_	161.0776	117.0551[M-H-CO_2_]^−^	4-Hydroxycoumarin ^b^	V2
11	13.233	283	C_7_H_6_O_2_	121.0277		m-Hydroxybenzaldehyde ^ab^	K1
12	14.701	284	C_7_H_6_O_2_	121.0295		p-Hydroxybenzaldehyde ^ab^	S1, S2, V2
15	16.83	287	C_8_H_10_O_3_	153.0174	125.0286[M-H-CO]^−^	Hydroxytyrosol ^b^	K1, K2
					123.0128[M-H-CH_2_O]^−^		
17	18.357	287	C_8_H_10_O_3_	153.0203	125.0286[M-H-CO]^−^	Hydroxytyrosol ^b^	V2
					123.0128[M-H-CH_2_O]^−^		
23	23.01	283	C_14_H_8_O_4_	239.0894	195.1382[M-H-CO_2_]^−^	Alizarin ^a^	V1, V2
28	31.256	282	C_13_H_14_O_3_	217.1048	172.8927[M-H-CO_2_]^−^	EUPATORIOCHROMENE ^b^	V2
33	36.732	264	C_18_H_24_O_3_	287.15	269.1377[M-H-H_2_O]^−^	2-Hydroxyestradiol ^b^	K2
					227.1307,209.5160		
34	35.801	295	C_13_H_10_O_6_	261.1502	125.0975[M-H-C_7_H_4_O_3_]^−^	Maclurin ^a^	S1, S2
					187.0988		

S1: acid hydrolysis-bound phenolics of soybean; S2: alkaline hydrolysis-bound phenolics of soybean; V1: acid hydrolysis-bound phenolics of vicia faba; V2: alkaline hydrolysis-bound phenolics of vicia faba; K1: acid hydrolysis-bound phenolics of kidney bean; K2: alkaline hydrolysis-bound phenolics of kidney bean. ^a^ Compared with the literature. ^b^ Compared with MSn data, data bases, and/or characteristic UV spectra. ^c^ Compared with an authentic standard.

**Table 3 foods-09-01816-t003:** Quantitative results of bound phenolics by HPLC-ESI-QqQ-MS ^a^.

Compounds	Soybean (μg/g DW)		Vicia Faba (μg/g DW)		Kidney Bean (μg/g DW)	
	Acid Hydrolysis	Alkaline Hydrolysis	Acid Hydrolysis	Alkaline Hydrolysis	Acid Hydrolysis	Alkaline Hydrolysis
**Phenolic Acids**						
p-hydroxybenzoic acid	2.11 ± 0.02	2.01 ± 0.23	19.96 ± 0.42	20.74 ± 0.42	2.14 ± 0.14	0.20 ± 0.04
procatechuic acid	8.69 ± 0.02	9.87 ± 0.21	46.87 ± 0.13	31.58 ± 0.36	16.13 ± 1.11	7.83 ± 0.11
ferulic acid	0.67 ± 0.03	0.17 ± 0.09	1.68 ± 0.10	1.07 ± 0.09	0.96 ± 0.12	0.19 ± 0.03
chlorogenic acid	Nd	0.84 ± 0.05	Nd	Nd	Nd	Nd
sinapic acid	0.16 ± 0.01	0.03 ± 0.01	0.25 ± 0.04	0.26 ± 0.01	0.20 ± 0.04	0.08 ± 0.03
gallic acid	Nd	Nd	0.32 ± 0.01	Nd	18.58 ± 0.68	9.57 ± 0.10
p-coumaric acid	3.09 ± 0.07	3.84 ± 0.15	0.66 ± 0.01	0.74 ± 0.01	2.13 ± 0.08	0.80 ± 0.02
**Flavonoids**						
Isoflavones						
daidzein	0.18 ± 0.01	nd	0.01	nd	nd	nd
daidzin	3.54 ± 0.01	3.99 ± 0.16	nd	nd	nd	nd
genistin	4.17 ± 0.06	3.87 ± 0.05	nd	nd	nd	nd
glycitin	Nd	0.82 ± 0.03	nd	nd	nd	nd
Flavones						
quercetin	Nd	0.18 ± 0.01	4.31 ± 0.07	1.79 ± 0.03	0.20 ± 0.02	0.33 ± 0.01
hyperoside	Nd	Nd	nd	nd	nd	nd
rutin	Nd	0.28 ± 0.14	nd	0.20 ± 0.01	nd	1.30 ± 0.02
vitexin	0.01	0.06 ± 0.01	0.02	nd	0.06 ± 0.01	0.05 ± 0.01
Flavanones						
naringenin	0.07 ± 0.01	0.02 ± 0.01	0.01	nd	nd	0.01
Flavanes						
catechin	Nd	Nd	7.97 ± 0.40	17.54 ± 0.22	0.84 ± 0.02	51.59 ± 1.18
**Total**	22.68 ± 0.24	25.98 ± 1.15	82.06 ± 0.82	73.92 ± 1.15	41.24 ± 2.22	71.95 ± 1.55

^a^ Results are expressed as mean ± standard deviation of three replicates.

**Table 4 foods-09-01816-t004:** Quantitative results of the phenolics after digestion by HPLC-ESI-QqQ-MS ^a.^

Compounds	Soybean (μg/g DW)			Vicia Faba (μg/g DW)			Kidney Bean (μg/g DW)		
	Oral	Gastric	Intestinal	Oral	Gastric	Intestinal	Oral	Gastric	Intestinal
Phenolic acids									
p-hydroxybenz-oic acid	0.01 ± 0.01	0.05 ± 0.01	0.16 ± 0.02	0.29 ± 0.09	0.54 ± 0.33	0.69 ± 0.02	0.61 ± 0.06	0.72 ± 0.10	1.22 ± 0.19
procatechuic acid	Nd	Nd	Nd	0.16 ± 0.02	0.22 ± 0.03	0.10 ± 0.01	0.40 ± 0.04	0.78 ± 0.11	1.28 ± 0.02
ferulic acid	Nd	Nd	Nd	nd	nd	nd	1.08 ± 0.03	2.49 ± 0.32	2.09 ± 0.23
chlorogenic acid	0.13 ± 0.01	0.12 ± 0.01	0.16 ± 0.07	0.19 ± 0.04	0.14 ± 0.02	0.20 ± 0.04	nd	nd	0.11 ± 0.02
sinapic acid	nd	nd	nd	Nd	nd	nd	0.01 ± 0.01	0.03 ± 0.01	0.06 ± 0.01
gallic aicd	nd	nd	nd	Nd	nd	nd	nd	nd	nd
p-coumaric acid	0.08 ± 0.01	0.09 ± 0.01	0.10 ± 0.01	0.50 ± 0.05	0.81 ± 0.01	0.34 ± 0.01	0.08 ± 0.01	0.08 ± 0.01	0.10 ± 0.01
Flavonoids									
Isoflavones									
daidzein	0.17 ± 0.05	0.30 ± 0.12	0.46 ± 0.04	nd	nd	nd	nd	nd	nd
daidzin	0.80 ± 0.02	0.85 ± 0.20	1.10 ± 0.06	0.04 ± 0.01	0.04 ± 0.01	0.06 ± 0.01	nd	nd	nd
genistin	0.71 ± 0.03	0.99 ± 0.12	1.26 ± 0.10	nd	nd	nd	nd	nd	nd
glycitein	0.08 ± 0.06	0.31 ± 0.04	0.44 ± 0.03	nd	nd	nd	nd	nd	nd
glycitin	0.09 ± 0.02	0.06 ± 0.01	0.06 ± 0.01	nd	nd	nd	nd	nd	nd
Flavones									
quercetin	nd	nd	nd	nd	nd	nd	nd	nd	0.13 ± 0.01
rutin	nd	nd	nd	nd	nd	nd	0.19 ± 0.04	0.09 ± 0.01	0.09 ± 0.01
vitexin	0.01 ± 0.01	0.02 ± 0.01	0.04 ± 0.01	0.004	0	0.009	nd	nd	nd
Flavanones									
naringenin	0.01 ± 0.01	0.01 ± 0.01	0.02 ± 0.01	nd	nd	nd	nd	nd	nd
Flavanes									
catechin	nd	nd	nd	0.90 ± 0.19	1.05 ± 0.32	3.66 ± 0.03	0.62 ± 0.04	0.21 ± 0.03	2.67 ± 0.09
**Total**	2.09 ± 0.23	2.80 ± 0.54	3.80 ± 0.36	2.08 ± 0.40	2.80 ± 0.72	5.05 ± 0.12	2.99 ± 0.23	4.40 ± 0.59	7.75 ± 0.59
Percent of acid hydrolysis/%	9.22	12.34	16.75	5.04	6.79	12.25	3.65	5.36	9.45
Percent of alkaline hydrolysis/%	8.04	10.78	14.63	3.25	4.38	7.9	4.04	5.95	10.48

^a^ Results are expressed as mean ± standard deviation of three replicates.

**Table 5 foods-09-01816-t005:** Total phenolic contents for different times of colonic fermentation (mg GAE/g DW) ^A^.

	0 h	1 h	3 h	6 h	12 h	24 h	36 h	48 h
Soybean	0 ^b^	1.43 ± 0.04 ^a^	1.50 ± 0.19 ^a^	1.64 ± 0.17 ^a^	1.49 ± 0.24 ^a^	1.72 ± 0.26 ^a^	1.30 ± 0.15 ^a^	1.65 ± 0.01 ^a^
Vicia faba	0 ^c^	0.85 ± 0.11 ^b^	0.93 ± 0.25 ^b^	0.91 ± 0.25 ^b^	1.16 ± 0.09 ^ab^	1.29 ± 0.28 ^ab^	1.48 ± 0.11 ^a^	1.45 ± 0.26 ^a^
Kidney bean	0 ^c^	0.73 ± 0.17 ^a^	0.40 ± 0.12 ^b^	0.56 ± 0.19 ^ab^	0.59 ± 0.06 ^ab^	0.79 ± 0.12 ^a^	0.82 ± 0.10 ^a^	0.81 ± 0.06 ^a^

^A^ Results are expressed as mean ± standard deviation of three replicates; Values followed by the different letters (a, b, c) within the same line are significantly different (*P <* 0.05).

**Table 6 foods-09-01816-t006:** Characterization of compounds during colonic fermentation by UPLC-ESI-QTOF-MS2.

	t_R_/	Formula	[M-H]^-^ (*m/z*)	Fragment Ions (*m/z*)	Identification	Fermentation Time			
	(min)					1 h	3 h	6 h	12 h
**Kidney Bean**									
1	4.635	C_14_H_12_O_5_	259.1302	241.1191,197.1293,171.1476	2,6,4′-trihydroxy-4-methoxybenzophenone ^b^			√	√
2	8.305	C_14_H_14_O_3_	229.154	185.1658	(Iso)pentenyl-7-hydroxy-coumarin ^b^		√	√	√
3	10.142	C_7_H_6_O_3_	137.0578	95.0489,122.0349	Hydroxybenzoic acid ^abc^		√	√	√
4	11.852	C_13_H_7_O_5_	243.1684	199.179	Xanthotoxol acette ^b^		√	√	√
5	13.82	C_13_H_7_O_5_	243.1651	199.1807,182.1586	Xanthotoxol acette ^b^		√	√	√
6	15.591	C_14_H_12_O_4_	243.17		Benzophenone ^b^			√	√
7	16.069		327.1298	291.1036	Brevifolin-carboxylic acid derivative ^ab^			√	√
8	17.986	C_10_H_12_O_4_	195.0653		Acetosyringone ^b^	√			
9	18.832	C_17_H_25_O_3_	277.1516		Decyloxy benzoic acid ^b^			√	√
10	44.099		391.2823	223.3473	Sinapic acid derivative ^ab^	√	√	√	
11	48.601		391.2902	343.2653	6-Hydroxy kaempferol 3,6,7-trimethyl ether derivative ^b^	√			
**Vicia Faba**									
1	5.277	C_9_H_10_O_2_	149.0689		4-ethylbenzoic acid ^b^			√	√
2	7.996	C_9_H_8_O_3_	163.1728		Hydroxyphenyl acrylate ^b^				√
3	8.001	C_9_H_10_O_4_	181.0481	163.0412,131.1925	Dihydroxybenzene propionic acid ^b^		√	√	√
4	8.099	C_14_H_14_O_3_	229.1533	185.1668	(Iso)pentenyl-7-hydroxy-coumarin ^b^		√	√	√
5	43.049		391.2808	223.3473	Sinapic acid derivative ^ab^	√	√		
6	48.614		391.2812	343.2653	6-Hydroxycalanol 3,6,7-trimethyl ether derivative ^b^		√		
**Soybean**									
1	3.079	C_16_H_10_O	217.1187	187.1087,130.0877	1-hydroxyindole ^b^	√	√	√	
2	8.103	C_14_H_14_O_3_	229.1593	185.0603	(Iso)pentenyl-7-hydroxy-coumarin ^b^		√	√	√
3	11.612	C_8_H_8_O_3_	151.0448	113.0778,109.1214	Hydroxyphenylacetic acid ^ab^				√
4	47.147	C_20_H_22_O_8_	389.266	435.2747,226.6650	Resveratrol glucoside ^ab^	□	√	√	√

^a^ Compared with the literature. ^b^ Compared with MSn data, databases, and/or characteristic UV spectra. ^c^ Compared with an authentic standard.

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
