# Peer review of "The Composition and Antioxidant Activity of Bound Phenolics in Three Legumes, and Their Metabolism and Bioaccessibility of Gastrointestinal Tract"

_foods, 2020, doi:10.3390/foods9121816_

Round 1

Reviewer 1 Report

The manuscript evaluates the chemical composition and antioxidant activity of bound phenolics of Glycine max, Phaseolus vulgaris and Vicia faba, and their gastrointestinal fermentation and digestion using in vitro models. I recommend it for publication after implementation and correction of certain data.

Below there are some specific comments/suggestions for its improvement.

Subsection Materials and Chemicals of the Materials and methods section: it is possible to suppose that seeds of the plants were analyzed, but it is not specified there. Mathematical rules should be applied for use of parenthesis e.g. [Glycine max (L.) Merr.].

Specify in details the manufacturers (company, city, country) and types of the all equipment and chemicals used. This information is completely missing for chemicals in the subsection Gastrointestinal digestion of bound phenolic of the Materials and methods section. Check also Thermo Varioskan Flash Microplate (page 5, line180).

When statement that “All experiments were performed independently in triplicate and the data were presented as mean value ± standard deviations (SD)” is provided in the Statistical analysis subsection of the Materials and methods section, the same information can be excluded from description of particular experiments (e.g. page 3, line 120 or page 5, line 216). However, meaning of term “independently” should be clarified.

In general, the structure of tables can be rationalized. For example, the first column of the Table 1 should be named “Antioxidative assay”, the second “Hydrolysis method” and categories in this column can be “acid” or “alkaline”. All abbreviations should be explained in the footnotes.

The detailed correction of a number of typographical errors and grammar of the whole text is strongly needed.

Reviewer 2 Report

Although it is a quality article, certain aspects must be improved:

  • the description of materials and chemicals should be exhaustive, as there are products that are not referenced in 2.1
  • item 2.4 must be described with the appropriate conditions.
  • being a scientific article with so much work, I think that the conclusion must be improved in order to enhance the bioavailability of the bound phenols, which is referred  in the article's objective.
  • English should be revised 
  •  Results and discussion in turn of result and discussion

Round 2

Reviewer 1 Report

No comments.